# Sustainable Development Goals and Gender Equality: A Social Design Approach on Gender-Based Violence

**Raquel Lima** [1] and **Graça Guedes** [2,*]

1   Facultat de Belles Arts, University of Barcelona, 08028 Barcelona, Spain; roliveli8@doct.ub.edu
2   Engineering School, University of Minho, Campus Azurém, 4804-533 Guimarães, Portugal
*   Correspondence: mgg@det.uminho.pt

**Abstract:** Gender equality is a central human aspect of the Sustainable Development Goals. Among its multiple and complex issues, this research highlights gender-based violence as a domain that affects women's empowerment and the guarantees of an effective equality on numerous levels. To address such a complex structure, which perpetuates inequalities between men and women, generating multiple effects and jeopardising social changes, social design can provide contributions on cultural and social levels. To achieve social systemic changes, one needs to activate profound cultural transformations. Thus, how can we change culture without rejecting the need to empower women and promote equality? The Montréal Design Declaration (2017) recognised social design's potential to achieve the Sustainable Development Goals (SDG), to contribute to global challenges, and to accept a calling for stakeholders' integration and agency promotion. This review explores how social design can provide contributions with regard to SDG5 and gender-based violence, presenting relevant domains that actively contribute to cultural transformation to address interventions in this systemic phenomenon.

**Keywords:** social design; gender equality; systemic changes; sustainable development goals





## 1. Introduction

Gender equality is probably the most relevant human pillar in the Sustainable Development Goals' central proposal. Because of its level of connection with the other SDGs and its large-scale effects on professional development, educational advancements, and the health and economic progression of individuals and nations, achieving gender equality is an urgent goal [1]. Gender equality means acknowledging that leaving half of our society behind will hinder the capacity to fight the great fight against global collapse. The world must support women's development in multiple areas to have many more specialists tackling the worst social and environmental problems. Otherwise, humanity will never achieve global transition.

However, more than legal mechanisms and public policies are needed to promote long-lasting change in this field due to its complex structure involving cultural beliefs and structural power dynamics [2]. It is essential to give women agency and to guarantee that they can have a presence and a voice on multiple levels, from private to public.

In addition to being a call to action, most SDG design interventions are silo solutions and social innovations from governmental, non-profit institutions, and entrepreneurs, diminishing the capacity to build systemic interferences in multiple contexts [3]. Thus, this paper provides theoretical evidence regarding the relevance of using a social design approach to enhance the articulation of the cohesion and solutions among gender equality issues. To show that it is possible to act from different perspectives, considering the complexity of the social issues [4], this study will focus on one of the most overwhelming effects of gender inequalities, which is gender-based violence (GBV), to explore possible social design territories that better fit the necessity to respond to complex issues and

to promote mechanisms for women's equality and agency. The research selected GBV because of its harmful effect on society, its multilevel structure comprising several actors, and because it includes third-sector institutions relevant in the field that act directly with victims and perpetrators [5]. In this context, social design will challenge complexity to enhance other well-known fields related to gender equality to fight against women's oppression, promoting the design approach and giving it broader grounds. Nevertheless, what worsens and perpetuates those inequalities is the difficulty involved in changing cultural patterns and norms [6] and the barrier to accepting changes. To transform cultural paradigms, confronting the current context and promoting new ways of thinking and acting is necessary. It involves the connection of many social and cultural instruments and dealing with subjective areas of human beings, such as vulnerability, biases, stereotypes, and so on [7]. This generates a central question: How can we propose a cultural change to promote equality and women's agency without rejection?

## 2. Gender Equality and Gender-Based Violence: The Foundation Stone

Gender inequalities disproportionately affect men and women in various areas, in both the private and public spheres. On the economic level, women and girls struggle for employment, and when they have the opportunity to work, the conditions are worse than they are for men. In education, despite the presence of women at all educational levels, this is not reflected in better wages, as women continue to receive up to 13% less than men [8]. Moreover, the gender gap in science, technology, engineering, and mathematics (STEM) represents a solid barrier to global development. This is also the case in politics, where, despite some advancements in parliamentary representativeness, the number of countries with a female head of state is low [9]. Women also struggle to access basic needs, such as water, soil and land, and health, which is reflected in the disease increase [10]. Finally, the persistence of violence against women has undermined their progress on many levels; this is worsened by COVID-19, climate change, and the current contexts of conflicts and migration [11]. The issues of gender inequalities are the result of hundreds of years of oppression against women. Additionally, the patriarchal and capitalist social functioning logic, which is reinforced by social and cultural mechanisms, profoundly affects women's personal development [12]. Because of the patriarchal structure, women have their individualities suppressed, generating psychological and segregating consequences, which are reflected in school and professional performances. In addition, historically, women did not have ownership over their bodies, serving men as productive and reproductive capital [13]. The effect of these structures that involve autonomy over bodies is still seen today in the regulations regarding reproductive rights in some countries, in the culture of beauty, and in stereotypes about female ageing.

These are just a few elements maintaining a patriarchal and homogeneously masculine system; these elements put women in unfavourable conditions compared to men, preventing their growth [8]. Hence, gender inequalities hinder almost half of the global population from pursuing their rights and dreams and their progress towards their full capabilities. In other words, blocking women's full development worsens the capacity to thrive and to overcome global collapse issues [1].

The 2023 Agenda for Sustainable Development Goals highlights the transdisciplinary effects of gender inequalities, making them a foundation for the achievement of the 2030 Agenda for Sustainable Development. The ultimate analytical document from the United Nations shows that women experience multiple forms of oppression on the seventeen levels of the SDGs [14].

The Women's Empowerment Index (WEI) and the Global Gender Parity Index (GGPI) are transparent: "No country has achieved high women's empowerment while maintaining a large gender gap. This suggests that women's and girls' empowerment will remain elusive until gender gaps are eliminated" (p. 1, [15]). Additionally, the report claims that higher human development is also not translated into women's empowerment. Empowerment

and gender equality are virtuous circles in which one receives positive reinforcement and actively propels positive transformations in another.

While the challenges are becoming more complex, their interrelations are making it even more challenging to intervene. Sachs [16] analysed the SDGs' interdependencies and elaborated six transformation points: (1) Education, gender, and inequality; (2) health, well-being, and demography; (3) energy decarbonization and sustainable industry; (4) sustainable food, land, water, and oceans; (5) sustainable cities and communities; and (6) digital revolution for sustainable development. Gender equality (SDG5) is evidenced in almost every transformation element, proving the centrality of this element at every point of intervention.

A call to action has been made since the 2023 Agenda and its new set of changes based on targets for prosperity, people, planet, and peace. It is an exercise to create consciousness about the ripple effects of some areas that undermine or evidence policy investments to achieve policy coherence; a systemic interconnection between multiple stakeholders is needed [17]. The research found potential SDG areas (1, 4, 11, 12, 14, 16) to improve SDG5 investments because approaching them will increase equality and other related areas for sustainable development [18].

The rise in complexity and the relevance of gender equality among the SDG pillars evidence the urgency of the deconstruction of inequalities against women. After all, the questions that are relied on but have no answers are: why has gender inequality been happening despite many efforts to improve this issue? Why, generation after generation, do we continue to see a prevalent unequal relation between men and women?

The Women's Empowerment Index (WEI) and the Global Gender Parity Index (GGPI) tools show that there is a need to balance the distribution of power so that women have full conditions in which to exercise their citizenship and their lives in general [19].

However, regardless of the area where gender inequalities operate, the stereotypes against women and the rejection of change persist. Research evidence shows that 9 out of 10 people have some prejudice against women and believe that men play a better role as political leaders and business executives [20]. It demonstrates how challenging oppressive cultural norms is necessary to change harmful social behaviour. A normalisation of women's vulnerability during life persists. Furthermore, common sense continues to assert that women need to be saved. In agreement with Fatou Wurie, empowering someone puts another person in a position of vulnerability [21], and this comes from a place of subjugation, particularly with regard to women. Rather than be saved, women need tools and support to find their agency as humans that thrive.

The biggest paradox in SDGs is far from implementation, but it is related to the acceptance and cultural transformation of every sustainable goal [22]. It relies on the cultural discourse and its resonance in the multi-context in which the interconnections of multiple SDGs coexist. There must be a transition from concept to acceptance and application for consistent changes. Thus, an integrated and self-managed society with correct sources and tools will be able to cross the cultural acceptance barriers [16]. However, there is no way to have one culture, as this could signify the death of other cultures. Hence, the understanding of multiple contexts and the ability to intervene is central to sustainable transformation.

Thus, the strategies and action plans that focus on behavioural changes in the sociocultural paradigm must be thought about in terms of long-term action, with a perspective of continuous learning and exercise because all the revisions and adjustments in international policy proposals and demands for a more significant female presence in different environments will not help if this paradigm shift is not accepted. In this regard, greater awareness, accountability, and civic action must exist.

*Contextualizing Gender-Based Violence*

As a consequence of gender inequality, gender-based violence (GBV) has its root and mental models based on social, historical, and cultural structures; this is aggravated by

the fact that it affects the vast majority of women, regardless of age, class, race, or origin. However, it affects the historically oppressed more strongly. Therefore, this section will be aimed at providing a brief overview of the general scenario of GBV, in order to contextualize the social design potential to systemically tackle the problem.

Even though the Declaration of Human Rights guarantees equal rights regardless of gender, it is necessary to consider gender in its multiple spheres as a performative and cultural attribution, including gender identity and gender expressions. However, in a binary world, the prevalence of male and female attributes can exclude non-binary and trans people from this protective perspective. Without the intention of erasing or disregarding other expressions of gender but by considering the limited access to data in this sector, this research focuses mainly on violence against women as a main force that hinders the potential for half of the society to thrive.

It is essential to address gender issues, as many women suffer daily oppression simply because they are women or because they live in a position where they have been historically and culturally assigned to a social model of behaviour and subjugated by men. It is worth mentioning that violence against trans and non-binary women has been experiencing high rates of domestic and sexual violence in many countries, due to transphobia and prejudice. According to the review, improvements in public resources and health investments are not enough to fight the discriminatory behaviours that prevent those women from living their choices freely.

Violence can have many forms, but when related to gender, no social group, culture, race, or age qualifies or typifies the phenomenon. According to the United Nations, the end of gender-based violence is crucial for managing global crises and for thriving societies [1]. As a violation of human rights, violence against women and girls has been alarming since the pandemic. The efforts to mitigate the impacts of GBV are varied and come from legislation related to social movements. However, the mechanisms we know are ineffective against gender-based discrimination and other obstacles to gender equity [23]. Inequalities are rooted in social and cultural norms solidified in our society. Thus, transformations need social cohesion and commonality that can create flourishing social roles on the side of the oppressed. As quoted by Paulo Freire (1921–1997), "Washing one's hands of the conflict between the powerful and the powerless means to side with the powerful, not to be neutral".

Despite all the efforts to deconstruct these structures over the generations, the numbers do not appear to fall. According to the United Nations, one-quarter of women around the globe suffer some violence in their relationships. The numbers are significatively high among women aged 15–49 years, who represent 30% of those around the world living in some form of oppression during their life journeys [24]. This is a condition that impacts the development of their potential and affects them psychologically, physically, and emotionally, not to mention the fact that almost 40% of feminicide is committed in an intimate context [25].

In Europe, research conducted with 42,000 women showed that 55% of them had experienced sexual harassment at least once since age 15. Alarming cases also happen in the work environment, where 75% of women have been sexually harassed, primarily by their bosses, colleagues, or customers (32%) [26]. Sexual harassment, including in education and cyber-harassment, affects young women's participation in social life and in being politically active [27]. Additionally, incidents like homicide, sexual assault, and rape are commonly committed by intimate partners or members of their family, showing that even their houses are not safe places [28].

The consequences of any act of violence can be seen as physical but they are mostly seen in their psychological effects. However, the side effects trespass on the survivors' sphere, entailing the expenditure of money on prevention and health, impacting economic development for companies and countries, and interfering with social progression. Moreover, themes that seem far from this phenomenon, like climate change, have inferences related to violence against women. Recent research points out how climate change propels

violence against women and girls when they are forced to leave their lands or in water scarcity situations. Their vulnerability presents an emergent negative impact and a lack of supportive structures and initiative [29]. According to independent consultant Astrid Puentes Riaño, there are reports where women were victims of violence inside those institutions that were supposed to protect them and when the institutions reported those women as prisoners [30].

The field of domestic violence goes beyond areas and interests, entering the most diverse spheres and populating our days in newspapers, television news, and social media. What we are experiencing today reinforces the humanitarian character of the issue, which, in 1990, at the Conference on Human Rights (Vienna, Austria, 1993), brought discrimination and violence against women to the centre of the United Nations activities. It was in 1993 that the Declaration for the Elimination of Violence against Women brought the foundations and new commitments to member states; these were strengthened at the 4th Conference on Women's Rights (Beijing, China, 1995), which evidenced the discrimination and violence against women and placed them as being the result of constructions of social gender inequalities based on noticeable differences between the sexes [31].

The ecological perspective of gender-based violence, curated by Heise [32], visually presents the interrelated aspects and effects of the phenomenon in personal and environmental life. This approach helped to develop contributions regarding the primary prevention of violence against women, where the community has a vital role in promoting non-violent behaviours and positioning itself as an ally against public abuse [33]. Thus, mobilising civil society figures is a valuable way to achieve social transformation in this field, which is benefitted by social cohesion and by synergy among the other participants in this system.

With that said, a multi-angle observation of the phenomenon is crucial to situate its transversal and multi-linearity structure better. Thus, briefly, it makes some of the aspects that compound GBV visible by presenting central elements, ideas, and concepts from sociology, anthropology, psychology, and law.

From the sociological angle, the comparisons between men and women have remained prevalent and have naturalised men; lately, the sense of neutrality has been attributed to them. To women, the attributes of fragility and incompleteness are solved only in the presence of men. The gender scripts are heteronormative cultural practices based on a binary perspective of behaviours that settle inequalities [34]. The dominance over women's bodies promotes continuous segregation and oppression, reproducing legitimation among male acts, whether manifested or institutionalized [34].

Through the decades, gender roles have shaped society, defining spaces, places, and behaviours. Lindsey [35] states that socialisation is central to societal disparity structures. To the author, it is not hormones that could be responsible for acts of aggression, as this is unclear and well proved, but "The cultural features of the context are powerful forces in determinants of aggression (p. 45, [35])".

Additionally, masculinity practice has been reinforcing violence. Depending on what masculinity is performed, violence is sometimes accepted and even celebrated, which legitimises and reproduces dominance [34]. For men, gender norms are more rigid in some ways, adding to the lack of emotional encouragement and to men being unable to express insecurities and being constantly pushed to be aggressive and to receive social approval.

From a psychological perspective, the China Convention has shed light on the impact on women's lives through violence, which can decrease their self-esteem and confidence, acting powerfully on their health [9]. Despite the evolving methods of psychological support for victims, COVID-19 was able to show the fragile system of interventions and the lack of mental support. Exposed to their abusive partners and obligated to stay at home, many women faced solitude, the incapacity to find help, and a latent rise in injury risk. Additionally, with unemployment and instability, women were more exposed to relational conflicts [36].

According to Walby [37], gender-based violence directly affects wages, absence, and work incapacity, reflecting productivity losses. However, it is to be added that costs can also be reflected in health systems, given the need to provide mental and physical support for victims. In addition, costs can also be reflected in the legal and criminal justice system, personal costs, social welfare, and other psychological costs. A case study estimated the total cost per year to be around EUR 13,732,068,214.00 [37].

From a legal perspective, it took many decades for the phrase 'Women's rights are human rights' to become a statement absorbed by law. The Declaration of Elimination of Violence Against Women, from 1993, was the first instrument to define and address forms of violence against women. It was consolidated in 1995 with the Beijing Declaration and Platform for Action, which settled 12 critical areas in which to advance women's rights [38]. The UN Millennium Declaration also set goals for 15 years, whereas the gaps and barriers could be addressed by the agenda of the Sustainable Development Goals [28].

Despite advancements in law and politics, the legal system is made by people with their own biases and beliefs, influencing results and advancements in this field. In many places worldwide, judges are not expected to consider offenders not guilty of domestic violence, elevating the perception of impunity [39]. Moreover, a legal apparatus is only sufficient for treating the root causes hidden in our culture and the elements perpetuating the status quo.

## 3. Social Design: The Genesis of Social Change by Design

Social design is systemic in its proposal and collaborative and participatory in its practices in providing solutions for and with people and communities. An idealistic proposal evolved into a practical and critical approach [40]. Social design proposes an interconnection of fields, learners, and actors as a facilitator to improve people's lives. In this session, the research navigates the multiple levels of social design that distinguish it from other material areas of this field.

Alongside the many worlds of design, there is a world that addresses the social challenges that were added as a lexicon in 2007, in addition to the urge for social responsibility that started in the 1970s [41]. This is a design that challenges the consumer norm of capitalist production and attempts to propose interventions into the original designations of design by satisfying the intrinsic human needs that are addressed for those of marginalised groups. As a social practice, its origins are close to those of social service workers inspired by the ecological theories and collaborative approaches that require a multilevel perspective [42]. Its practice is correlated with methodological terms and, for many, can also be redundant to social design when everything design produces should be social [43].

It is interesting to note that the very evolution of design, which is well documented by Buchanan in his book on rhetoric, also follows the movements that sciences and global thinking developed from knowledge [44]. The construction of a design discipline takes place in a boiling cauldron of ideas, ideals, and great schools, like Bauhaus and Ulm, which positioned themselves worldwide and designed a much more responsive design with the ability to include other non-designers in their explorations and to plan applications and measurements. However, after the post-industrial hangover and the need to rescue the "soul of design", the contribution of machines to making art and fundamental human desires became a question, questioning the consumerism and injustices resulting from excessive materialism [44].

According to Koskinen and Hush [45], social design concepts can have two areas: molecular social design and sociological social design. The first focused on punctual social transformations, and the second on sociological theories articulated to generate answers for social inequalities. While molecular social design relies on a rational way of thinking that avoids utopian standards, in sociological social design the theory is essential to provide a context of intervention and ground action, positioning design as a critical actor. However, as affirmed by Weber [46], there is a challenge in sociological social design related to its outcomes being distant answers to its profoundly critical point of view.

Some authors claim that social projects can only be considered social design projects if their impact exceeds the original destination and expands in scope and beneficiaries [47]. However, social design also has its local valences, and it influences the microcosm in which it operates, with a relevant impact [48]. Its influence on collective and subjective participation makes social design valuable for current transformations.

It combines intangible subtle aspects and relational uses, where creativity and purpose can affect many more than just those of a specific group. On the other hand, even when it produces tangible answers, it is not easy to identify its whole impact or collect all the pieces of evidence because social design connects many layers of structures on personal and institutional levels from a system perspective [49].

There are also considerations about the domain as a design-based practice or a social intervention per se. The action research nature of social design and its collaborative proposal make this domain closer to social science [50]. On the other hand, its uses related to social innovation bring it into the entrepreneurial context, where designers can intervene in new ways of social dynamics rather than be the central focus of social responsibility in what is created [51].

As Gaitto [52] states, design is a tool of the economy in a symbiotic exchange and, consequently, is directly linked to the politics of the context in which it operates in its broadest form. Additionally, social design contributes to design policies, using the participation of society, helping it to redesign social systems while generating innovations that include citizens and propel resilience [53].

To Margolin (2015), the role of social design is to be an agent that organises multiple interventions in the global action for good, which can be fully seen in works from the global south communities, in which ground forces, citizens, and experts join forces to intervene in local social problems [54]. From another perspective, the Euro-centrality and Anglo-centrality of design reinforce the colonial prevalence of the saviour proposal, which is a dominant position per se [55].

Thus, design in socially oriented contexts assumes a supporting role. For this, it is also necessary to deconstruct the sovereignty and supremacy of what one does, making room for an exercise in humility and vulnerability as tools of self-awareness and empathy [54]. It is the work of social design in a place where the 'white saviour' complex gives way to a horizontality of knowledge sharing in which the designer is the apprentice of his interlocutors, and both lead the ecosystem towards change [56].

The design initiated as a discipline intended to solve problems through the physicality of its artefacts now occupies a more subjective purpose—as an articulator, connector, and coadjuvant in social transformations, providing tools to create dialogues between actors that, until then, they did not articulate [57]. It is not just about methodological proposals, as exchanges and sharing in collaborative fields are also the domain of other areas, such as sociology and anthropology [58]. What makes the design unique is its capacity to create visual and subjective dialogues that give its interlocutors the capacity for action and empowerment [59].

In this sense, social design cannot be merely an area of application of human-centred collaborative methodologies but rather an aggregator, facilitator, and connector of knowledge that is shared [60], where the moments of sharing are for the benefit of those who use them and not for the design, as the use of its tools and processes is knowledge that will belong to the other, as a way of empowering and improving the work of the collective [61].

Recently, Cheryl Heller [43] presented a broad representation of what social design can do and its impacts, contributing to an amplified view of the action component in this field. Despite social design being infused with social innovation, Prichard [62] defines it as "the design of the invisible dynamics and relationships that affect society and the future. It creates new social conditions intended to increase human agency, creativity, equity, resilience, and our connection to nature (p. 2)".

There is not only a definitive definition of social design. There are also some limits under this design field: "Contemporary design practices are mainly construed to support

creating objects, interactive devices, spaces and intelligent systems, but these practices give designers little help in the area of abstract social entities and how to work with them" (p. 3, [48]).

This amplitude of interventions highlights some essential areas where social design has powerful relevance and where it can see its impacts on society. Furthermore, the design's critical, deliberative, and political character is evidenced in six rising disciplines: social innovation design, pluriversal design, dialogical design, systemic design, political design, and transition design as summarized in Table 1. These approaches clarify the applications and outcomes of social design to extract evidence for its application to gender inequalities.

Few approaches are more effective in transforming mental models and behaviours than education [63], which allows the questioning and the construction of knowledge. No instrument is more effective in joining educational practice and critical thinking than social design [64]. This design area, which is dedicated to embracing human challenges, has gained attention in recent years because of its capacity to connect multiple actors and its tools to engage people in transformation. Thus, social design represents a perfect tool for addressing complex societal problems.

Social design tools and methodologies allow the development of a paradigm change at the personal and organisational levels [65]. Hence, social design finds fertile land to contribute to social transformation by using its creative and innovative capacity to transform people into social agents, promoting community empowerment and equity [66]. Its human-centeredness and participatory approach are proven tools with which to engage people in contexts where collaboration and co-creation are needed [67]. Moreover, rather than having a problem-solver mindset, design is driven by context, and from this perspective, adaptative thinking and learning enhance the need for resilience in tackling societal issues [43].

**Table 1.** Emergent design disciplines in the field of social design.

| Design Discipline | Explanation | Authors |
|---|---|---|
| Design for social Innovation | Social innovation became essential to move forward the conceptual idealisation to an authentic product of design for social change. Its outcomes are meaningful and based on new social and economic models. Additionally, the diffusion of design thinking as a model of approach was evidenced by social innovation and its entrepreneurial spheres and collective network proposal. | Margolin, V. and Margolin, S. 2002 [40]; Mulgan, G. et al., 2008 [48]; O. 2018 [65]; Phills, J., Deiglmeier, K. and Miller, D., 2008 [67]; M. Mortati, M. and Villari, B. 2014 [68]; Manzini, E. 2015 [69]; Deserti, A., Rizzo, F. and Cobanli, Manzini, E. 2018 [70]; Caetano, A. 2019 [50] |
| Dialogical design | Dialogical design helps to structure multiple ideas coherently using dialogues to capture and map causal systems. This ability to create dialogues between actors and between these actors and their surroundings is natural to human beings. Thus, dialogue is a central element in social design that joins consciousness, human approach and interactions, participants' relations, power dynamics, and emancipation. | Kimbell L. and Julier, J. 2012 [71]; Banathy, B. 2013 [72]; Cipolla C. and R. Bartholo, 2014 [57]; Irwin, T. 2015 [73]; Klumbytė, G. et al., 2022 [74]. |
| Transitions Design | The domains of transition design contribute to its practices, providing tools to approach system problems so their practitioners can visualise and intervene sharply in the field [68]. While in regenerative design, the wholeness and human nature propel the creation of solutions from a non-human perspective, and with a whole perspective interact with design culture, promoting a transformative aspect from aesthetics to mind models for positive emergences. | Irwin, T. 2015 [73]; Christakis, A. 1998 [75]; Du Plessis, C. 2012 [76]; Wahl, D. 2016 [77]; M. van der Bijl-Brouwer, 2017 [78]; Heller, C. 2018 [43]. |

**Table 1.** *Cont.*

| Design Discipline | Explanation | Authors |
| --- | --- | --- |
| Political Design or Design Justice | Political design creates ways for argument and for contesting the status quo, generating spaces and opportunities for debate and changing structures through its critical approach. Thus, one critical social design indicator is its impact and the level of autonomy such design intervention causes. In other words, how such interference shakes or changes the current reality for the better. | Fry, T. 2010 [79]; Freire, K. et al., 2011 [80]; Vazquez, R. 2017 [81]; Schultz T. et al., 2018 [82]; Costanza-Chock, S. 2018 [40]; Serpa, B. et al., 2020 [83]; Collins, P. 2015 [84]; Van Amstel, F. et al., 2022 [85]. |
| Systemic Design | In many fields, social innovation evolved to embrace the systems perspective, moving from a product and service provider to a complex service system view, including public participation and private and citizen representatives. Thus, systems thinking not only enriches social design practices but also propels the creation of more assertive and impactful proposals. Rather than a problem-solving perspective, the systemic design approach enables practitioners to navigate an exploratory journey for leverage points and emergencies that impact the whole. | B. Banathy, H. 2013 [72]; Christakis, A. N. 1998 [75]; Bertalanffy, von L. 1968 [86]; Meadows, D. 2008 [87]; Metcalf, G. 2014 [88]; van der Bijl-Brouwer M. and Malcolm, B. 2020 [89]. |
| Pluriversal and Regenerative Design | Pluriversal Design is embedded within many worlds. It allows a collective construction based on multiple voices, from humans and non-humans. Its principles enrich the design approach on multiple levels, from self-consciousness to collective management. These approaches enrich the design capability to answer social and environmental problems by accepting multiple voices, narratives, and truths. | Klumbytė G. et al., 2022 [74]; Wahl, D. 2016 [77]; Kania J. and Kramer, M. 2013 [90]; Escobar, A. 2015 [91]; Escobar, A. 2018 [92]; Noel, L.-A. 2020 [93]. |

These emerging disciplines that intersect with design for social change from multiple angles evidence the design practice capability to engage, transform, and evolve the human capacity to thrive. Inside these topics, it was interesting to find elements that gather together those different disciplines that could be explored in further investigations and practices in collective and systemic approaches; this is further explored in the following section.

## 4. The Six Social Design Domains for Social Change in GBV

All advancements in gender-based violence structures signify a step forward against women's oppression. However, cultural and social behaviour cannot follow juridical and mechanical evolution, making gender equality goals even more complex. Thus, fundamental and incremental innovation support must be applied to enable people and institutions to better and more rapidly develop practical impact. As Murray, Caulier-Grice, and Mulgan say, "Current policies and structures of government have tended to reinforce old rather than new models. The silos of government departments are poorly suited to tackling complex problems which cut across sectors and nation-states. Civil society lacks the capital, skills and resources to take promising ideas to scale (p. 4, [69])".

Hence, multilevel interventions that systemically address the issues of violence against women and girls are essential to advance and use the participatory approaches responsible for critical thinking and engagement for a long-lasting intervention. All these aspects provide clear evidence about the relevance of social design contributions to gender equality, particularly with regard to GBV.

With the intention of clarifying the levels of change in gender-based violence and gender equality and the cultural and social beliefs that they rely on as their root causes, Figures 1 and 2 show an evolving representation of cultural transformation and participation. However, transformations take time, practice, and adaptation to be effective, as

represented in Figure 1. At the same time, as a sublayer, there is an individual and collective force that must be conscious and engaged, as demonstrated in Figure 2.

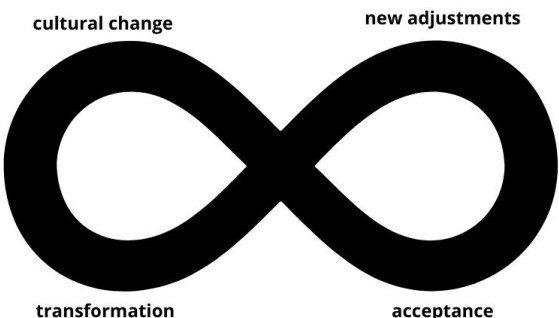

**Figure 1.** Illustration of cultural and social transformation.

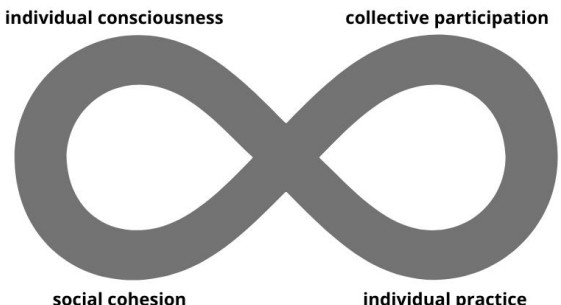

**Figure 2.** Illustration of collective and individual participation.

Despite the lack of design responses in GBV as well as in the Sustainable Development Goals, there is research that evidences how design can enhance the human capability to accept change and to experience belonging in order to tackle societal issues. Those practice-based approaches can show the tremendous potential of design in social change; this is further discussed through the six domains that correspond to a suggested focus of implementation.

Regarding the finding of mechanisms capable of elevating social change in GBV, which has a prevalent need for cultural transformation and acceptance, the literature review provided a set of territories that could directly impact decision makers, policy makers, and designers to address the approach to the mindset and to corroborate to propel the use of social design to tackle GBV and gender equality-related issues.

Thus, the cultural and social aspects for social change are movements of practice and acceptance, where social design can contribute with its participative, systemic, and critical tools and interventions.

To suit this complex approach, there are six central domains where social design can create change in GBV and its interrelated structures. Through these points, GBV can be addressed in relation to structural, cultural, and behavioural spheres, multiple prevention mechanisms, different contexts and realities, multiple stakeholders and causal structures, and evolving synergies and commitment. They are considered a baseline for social design intervention in gender equality due to their nature and their activation of mental model changes through their practices.

1. Systemic discipline

Its multiple lenses make the dynamics visible and its participants aware of their roles and the disturbances or emergencies they cause, whether intended or not. Enabling people to see the system helps them choose the right solutions better, see others' needs, exchange resources, and plan accurate strategies. As they see the system, they can embrace responsibility for the current reality, making decisions based on multiple effects and avoiding reactive orientation among the problems.

A system approach is ideal for decision makers to tackle social problems because it enables multiple participants to be aware of their roles and current impacts on the system. This vision leads to better choices and frame leverage points that will consequently make long-lasting contributions.

2. Critical practice

Social design (SD) contributes to the evidence on the critical character of design for social change. It gives visibility and voice to social issues, allowing them to be explored in depth by multiple actors and with community participation. Its democratic being is a call for critical thinking and social engagement in practical ways. These practices can become new cultures of social action and intervention. Moreover, as a participatory methodology, it provides a collective capacity for reflection in action.

A critique of design, in general, is directly reflected in social design, which is about its ability to intervene in social and cultural fields. The culture capital or social capital is the combination of mental and behavioural structures that self-regulate society. Thus, everything that intervenes in social or cultural domains is subject to moral and ethical questioning. It is what has been conducted in the multiple ways and domains of design knowledge, such as design justice, design of oppression, decolonial design, and design activism.

3. Conscious practice

Social design not only makes the structures visible but also propels self-awareness. The conscious mindset is essential to promote social change. It is also a potential tool to elevate a socially responsible design path by evidencing its benefits as a responsive design. This can also be described as inclusive design when multiple methods are incorporated to achieve inclusive outcomes that help to diminish inequality gaps. Hence, acknowledging and accepting different perspectives is the awareness of contexts and their implications in the design process for sustainable futures.

4. Emancipatory tool

Social design acts from a supportive angle, giving practitioners the right tools to enable change. Additionally, it sees everyone as a designer and an individual capable of generating positive transformations. The starting point is to nurture people to give their best, and the result is often that the participants are engaged and committed to those transformations. As an emancipatory tool, SD embraces different cultures and contexts, adapting its proposal through an emphatic approach following areas such as participant and research safety, ethical frameworks, and respect and understanding of cultural domains.

5. Relational tool

The relational domain in social design enhances the capacity to connect multiple actors and the underlying structures of human interrelations. Its tools and approach to deconstructing power dynamics are crucial in sensitive social domains, helping to create and maintain partnership and collaboration. Moreover, its participatory approach enables all voices to be heard and all presences to be represented.

Practical evidence of the benefits of dialogic interventions is the work with victims using their memories for trauma release. For example, the work uses a participatory approach and theatre to empower women and the gender-sensitive approach. Moreover, social design creates spaces for social cohesion, a form of inclusion that allows culturally diverse perspectives.

6. Support for transitions

The siloed thinking and cause–effect solutions that drive the current context in GBV can benefit from a social design approach for transitions, which enables institutions to face complex problems in a broad and co-creative way. Throughout the principles of design transitions, practitioners can visualise interconnectedness and interdependencies, bridge stakeholders, and co-create with them to identify the best intervention points. As other areas have benefited from this sustainable vision proposed by transitions, GBV can also

envision new ways of being approached collectively, activating the total capacity to achieve a sustainable and more inclusive future for all.

These approaches, illustrated in Figures 3 and 4, show social design in each phase of social transformation, where one domain can manifest with more emphasis than others, as displayed in Figures 3 and 4. These suggested domains can improve social transition in cultural change and the acceptance of new paradigms in gender equality. It can be an initial design approach that outlines every phase of the transformation of cultural change, whether it is applied in the education fields of gender equality and gender-based violence or at the institutional levels. It considers cultural change as a long-term journey of commitment and evolving transformations. Thus, it needs a proposal capable of helping in the navigation through moments and in adaptation with a resilient focus.

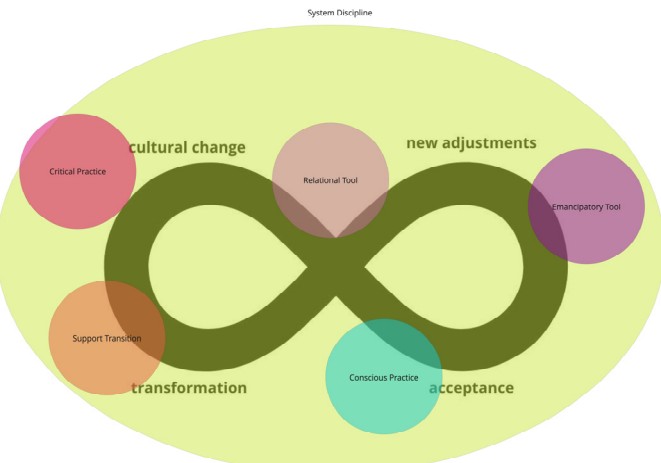

**Figure 3.** Social design baseline interventions in social transformations.

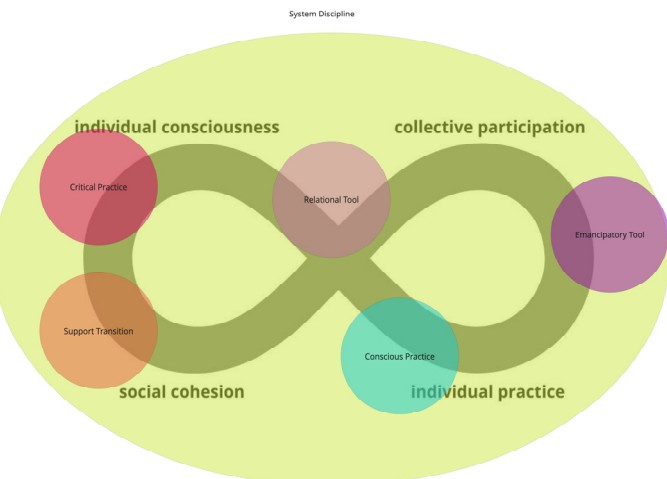

**Figure 4.** Social design baseline interventions in individual participation.

## 5. Conclusions

In a fragmented society willing to achieve sustainable goals, it is essential to propel conviviality, partnership, and solidarity dynamics. Hence, it relies upon the potential to enable change; social design applied to help decision-makers acts as a ripple tool to generate transformation in gender equality and violence against women and girls. Its potential to connect people and ideas and to add multiple worlds into the conversation makes new ways of collaboration, affection, self-awareness, and presence in life flourish. Social design's experimental and experiential nature allows system actors to actively feel, understand, and react, gaining self-consciousness and accountability. Moreover, it needs to

make vivid the design idea of human betterment as a principle of design, where humanity uses its full potential to transform its reality, enhancing networks and enabling them to act in a manner which is conscious, systemic, relational, supportive, emancipatory, and critical.

**Author Contributions:** Conceptualization, R.L. and G.G.; methodology, G.G.; validation, G.G.; formal analysis, G.G.; investigation, R.L.; resources, G.G. and R.L.; data curation, G.G.; writing—original, R.L.; draft preparation, R.L.; writing—review and editing, G.G.; supervision, G.G.; project administration, G.G. and R.L. All authors have read and agreed to the published version of the manuscript.

**Funding:** This research was funded by FCT–Foundation for Science and Technology, I.P., under the project UID/CTM/00264/2020 of Centre for Textile Science and Technology (2C2T).

**Institutional Review Board Statement:** Not applicable.

**Informed Consent Statement:** Not applicable.

**Data Availability Statement:** No new data were created or analyzed in this study. Data sharing is not applicable to this article.

**Conflicts of Interest:** The authors declare no conflicts of interest.

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
