# Peer review of "Sustainable Development Goals and Gender Equality: A Social Design Approach on Gender-Based Violence"

_sustainability, doi:10.3390/su16020914_

Round 1
Reviewer 1 Report
Comments and Suggestions for Authors
The view “Sustainable Development Goals and Gender Equality: A Social Design Approach” , deals with a very important and up-to-date problem of gender inequalities which disproportionately affect men and women in various areas, both in the private and public spheres, on the economic, educational and political level. The key issue is that the Sustainable Development Goals cannot be achieved without ensuring gender equality.
The paper is of a theoretical nature. The aim of the view is to explore the ability of social design to respond complex issues and promote mechanisms for women's equality and agency. The authors focused on one of the most overwhelming effects of gender inequalities, which is gender-based violence. They formulated the following research question: How can we propose a cultural change to promote equality and women's agency without rejection? As a result, they evidence six social design domains which can actively contribute to cultural transformation in gender equality issues.
Strengths of this study:
- the topic of the paper is very important,
- the paper is properly structured,
- the bibliographic review is exhaustive and up-to-date,
- the paper is easy to read.
My suggestions:
- it is recommended to rephrase the title of the paper formulated by authors because it does not indicate that the authors focused on gender-based violence,
- the purpose as well as the scope of the paper should be specified more precisely in the abstract,
- the authors could supplement the 6th part of the paper with a clear reference to the research question formulated in the introduction and by defining the prospects for further research.
In my opinion this view qualifies to be published in Sustainability with this minor modification.
Author Response
Thank you very much for your time reviewing these papers, we appreciate your comments and consideration. Please find the detailed responses below in the point-by-point response and the corresponding revisions/corrections highlighted/in track changes in the re-submitted files.
Point-by-point Response to Comments and Suggestions for Authors
Comments 1:
“it is recommended to rephrase the title of the paper formulated by authors because it does not indicate that the authors focused on gender-based violence”
Response 1:
Agree. We have, accordingly, changed: Sustainable Development Goals and Gender Equality: A Social Design Approach on Gender-based Violence
Comments 2:
“the purpose as well as the scope of the paper should be specified more precisely in the abstract”
Response 2:
Agree. We have improved the abstract to make it clearer, as can be seen in the re-submitted file, highlighting the focus on gender-based violence and the social design contribution to cultural transformations.
Comments 3:
“the authors could supplement the 6th part of the paper with a clear reference to the research question formulated in the introduction and by defining the prospects for further research.”
Response 3:
Agree. We made substantial changes in the 4th and 5th parts of the paper to evidence social design domains and their possible contributions, where we added figures to better answer the research question. You will find those changes from line 407, in Figures 2 and 3, and line 500, and Figures 4 and 5.

Reviewer 2 Report
Comments and Suggestions for Authors
The manuscript „Sustainable Development Goals and Gender Equality: A Social Design Approach” is about a very important problem that was in the past but also is very visible in the contemporary world. However, the manuscript does not bring anything new. The Authors emphasize how important is gender equality and try to indicate six social design domains where social design can flourish change in gender-based violence. There is nothing new that the Authors show multidimensionality of gender equality and the need for treatment of this problem in many areas.
Besides, the manuscript is inconsistent. The ‘Social Design: The Genesis of Social Change by Design’ section is not related to the next section (‘Contextualizing Gender-based Violence’). The ‘Conclusion: The Six Social Design Domains for Social Change in GBV’ section being, in my opinion, the key section does not cover any research gap. In this section, the Authors tried to combine considerations from two previous sections.
I also have some detailed comments.
First of all, the title does not cover the manuscript content. The manuscript focuses on gender-based violence and this term is not used in the title of the paper but it is also not used in the abstract.
The numbering is incorrect. There is section 3, after this section again there is section 3 and the next section has number 6.
On page 6 the Authors write about ‘Eurocentrality and Alglocentrality’. I do not find the term ‘Alglocentrality’. I think that there is a typing error.
The key section ‘Conclusion: The Six Social Design Domains for Social Change in GBV’ should not be titled that way. The last paragraph of the text should be a conclusion and the rest of this section should be titled ‘The Six Social Design Domains for Social Change in GBV’.
There are some errors in the references (e.g. in the 49th reference there is an error in the publication title).
Additionally, the abbreviation SDG is used without earlier explanation (p. 1).
Comments on the Quality of English LanguageThe Authors often use uppercase and lowercase letters for the same terms, e.g. ‘gender-based violence’ and ‘Gender-based Violence’, ‘Gender equality’ and ‘gender equality’. This problem is related to many terms.
Author Response
Thank you very much for your time reviewing these papers, we appreciate your comments and consideration. Please find the detailed responses below in the point-by-point response and the corresponding revisions/corrections highlighted/in track changes in the re-submitted files.
- Point-by-point response to Comments and Suggestions for Authors
Comments 1:
The manuscript „Sustainable Development Goals and Gender Equality: A Social Design Approach” is about a very important problem that was in the past but also is very visible in the contemporary world. However, the manuscript does not bring anything new. The Authors emphasize how important is gender equality and try to indicate six social design domains where social design can flourish change in gender-based violence. There is nothing new that the Authors show multidimensionality of gender equality and the need for treatment of this problem in many areas.
Response 1:
The theme is not new, however, a theoretical exploration that evidences the cultural aspects that structure the phenomenon is central to defining it as a field for design intervention. Unfortunately, the design did not explore gender equality issues that much, and the same happens with gender-based violence and violence against women. Hence, this manuscript intends to proclaim further explorations from design in changing cultural structures using its domains and disciplines.
Comments 2:
Besides, the manuscript is inconsistent. The ‘Social Design: The Genesis of Social Change by Design’ section is not related to the next section (‘Contextualizing Gender-based Violence’). The ‘Conclusion: The Six Social Design Domains for Social Change in GBV’ section being, in my opinion, the key section does not cover any research gap. In this section, the Authors tried to combine considerations from two previous sections.
Response 2:
We agree that these sessions can be clearer and we have, accordingly, made some arrangements in structure and bringing together the contextualization. You find some changes in line 370 and Figure 1, that underline the main theoretical approach from social design. We also made substantial changes in the 4th and 5th parts of the paper to evidence social design domains and their possible contributions, where we added figures to better answer the research question. You will find those changes from line 407, and in Figures 2 and 3, and in from line 500, and in Figures 4 and 5. In this form, we intend to improve the flow and the intersections with chapters evidencing possible contributions that answer the main question in the paper.
Comments 3:
First of all, the title does not cover the manuscript content. The manuscript focuses on gender-based violence and this term is not used in the title of the paper but it is also not used in the abstract.
Response 3:
Agree. We have, accordingly, changed: Sustainable Development Goals and Gender Equality: A Social Design Approach on Gender-based Violence
Comments 4:
The numbering is incorrect. There is section 3 after this section again there is section 3 and the next section has number 6.
Response 4:
Agree. A proper alteration was made.
Comments 5:
On page 6 the Authors write about ‘Eurocentrality and Alglocentrality’. I do not find the term ‘Alglocentrality’. I think that there is a typing error.
Response 5:
Agree. It was a type error and has been corrected.
Comments 6:
The key section ‘Conclusion: The Six Social Design Domains for Social Change in GBV’ should not be titled that way. The last paragraph of the text should be a conclusion and the rest of this section should be titled ‘The Six Social Design Domains for Social Change in GBV’.
Response 6:
Agree. The session The Six Social Design Domains for Social Change in GBV is separated from the conclusion and some incremental information was added with Figures 2 and 3, and Figures 4 and 5.
Comments 7:
There are some errors in the references (e.g. in the 49th reference there is an error in the publication title).
Response 7:
Agree. We corrected and reviewed the references.
Comments 8:
Additionally, the abbreviation SDG is used without earlier explanation (p. 1).
Response 8:
We understand that explaining Sustainable Development Goals in a journal that focuses specifically on Sustainable Development Goals could sound repetitive and will not add that much, thus we made better use of focusing on SDG5 and gender equality to better contextualize our points.

Reviewer 3 Report
Comments and Suggestions for Authors
I like the theme of this paper. I have two main concerns.
1. You use repeatedly the term "gender" but your discussion is only upon women. Current scholarship needs to reflect a more complex understanding of gender than a binary male/female. I think this needs to be explained and discussed, and then you may state that you choose to focus only on women (I suspect you are excluding transwomen, you should be clear on this).
2. You spend an enormous amount of space discussing design literature and then gender-based violence as concepts. Only a small portion of the manuscript focuses upon the intersection of the two, within the SDG. I strongly suggest a greater focus upon what your title promises, a discussion of that intersection AND some examples of application. This will require a thinning of the literature discussion, but I think that is critical for this manuscript: the intersection and application.
3. Proofread, there are some incomplete or overly complicated sentence structures.
Comments on the Quality of English LanguageAcceptable, but proofread.
Author Response
Thank you very much for your time reviewing these papers, we appreciate your comments and consideration. Please find the detailed responses below in the point-by-point response and the corresponding revisions/corrections highlighted/in track changes in the re-submitted files.
- Point-by-point response to Comments and Suggestions for Authors
Comments 1:
You use repeatedly the term "gender" but your discussion is only about women. Current scholarship needs to reflect a more complex understanding of gender than a binary male/female. I think this needs to be explained and discussed, and then you may state that you choose to focus only on women (I suspect you are excluding transwomen, you should be clear on this).
Response 1:
Thank you for pointing this out. It is true that this proposal explores gender inequalities and doesn´t go further on gender discussions centred on the effects of SDG. Therefore, we provide an introductory context to state our focus on gender-based violence against women specifically or with more emphasis on women. You can find these adding from lines 153 to 168.
Comments 2:
You spend an enormous amount of space discussing design literature and then gender-based violence as concept. Only a small portion of the manuscript focuses upon the intersection of the two, within the SDG. I strongly suggest a greater focus on what your title promises, a discussion of that intersection AND some examples of application. This will require a thinning of the literature discussion, but I think that is critical for this manuscript: the intersection and application.
Response 2:
Agree. Unfortunately, this field is not exhaustively explored by design, but I provided information to activate design´s capacity to generate answers in sociocultural phenomena such as violence against women. we have, accordingly, made some arrangements in structure and bringing together the contextualization. You find some changes in line 370 and in Figure 1, that underline the main theoretical approach from social design. We also made substantial changes in the 4th and 5th parts of the paper to evidence social design domains and their possible contributions, where we added figures to better answer the research question. You will find those changes from line 407, and in Figures 2 and 3, and line 500, and Figures 4 and 5. Moreover, we lessen the literature discussion evidencing the macro themes, as you suggested. In this form, we intend to improve the flow and the intersections with chapters evidencing possible contributions that answer the main question in the paper.
Comments 3:
Proofread, there are some incomplete or overly complicated sentence structures.
Response 3:
Agree. Proofread was provided as well as some arrangements to clear the paper.

Round 2
Reviewer 2 Report
Comments and Suggestions for Authors
The Authors have improved their manuscript. Now, the text is definitely better, the structure of the text is also correct. The title of the paper correctly shows the content of the manuscript. However, I still have some doubts about filling some research gap but my concerns are now smaller. I am waiting for further Authors' explorations in this field.
Comments on the Quality of English LanguageI have no comments.
Reviewer 3 Report
Comments and Suggestions for Authors
I appreciate your revision efforts.